# TOPOLOGICAL AUTOENCODERS

## ABSTRACT

We propose a novel approach for preserving topological structures of the input space in latent representations of autoencoders. Using *persistent homology*, a technique from topological data analysis, we calculate topological signatures of both the input and latent space to derive a topological loss term. Under weak theoretical assumptions, we can construct this loss in a differentiable manner, such that the encoding learns to retain multi-scale connectivity information. We show that our approach is theoretically well-founded and that it exhibits favourable latent representations on a synthetic manifold as well as on real-world image data sets, while preserving low reconstruction errors.

## 1 INTRODUCTION

While topological features, in particular multi-scale features derived from persistent homology, have seen increasing usage in the machine learning community (Carrière et al., 2019; Guss & Salakhutdinov, 2018; Hofer et al., 2017; 2019a;b; Ramamurthy et al., 2019; Reininghaus et al., 2015; Rieck et al., 2019a;b), using topology *directly* as a constraint for current deep learning methods remains a challenge. This is due to the inherently discrete nature of these computations, making backpropagation through the computation of topological signatures immensely difficult or only possible in certain special circumstances (Chen et al., 2019; Hofer et al., 2019b; Poulenard et al., 2018).

In this work, we present a novel approach that permits obtaining gradients during the computation of topological signatures. This permits employing topological constraints while training deep neural networks as well as building topology-preserving autoencoders based on the following contributions:

1. We develop a novel topological loss term for autoencoders that helps harmonise the topology of the data space and the topology of the latent space.
2. We prove that our approach is stable on the level of mini-batches, resulting in suitable approximations of the persistent homology of a data set.
3. We empirically demonstrate that our novel loss term aids in dimensionality reduction by preserving topological structures in data sets; in particular, the learned latent representations are useful in that the preservation of topological structures can aid interpretability.

## 2 BACKGROUND: PERSISTENT HOMOLOGY

Persistent homology (Barannikov, 1994; Edelsbrunner & Harer, 2008) is a method from the field of computational topology, which develops tools for analysing topological features (connectivity-based features such as connected components) of data sets. We first need to introduce the underlying concept of simplicial homology. For a simplicial complex $\mathfrak{K}$, i.e. a generalised graph with higher-order connectivity information such as cliques, simplicial homology employs matrix reduction algorithms to assign $\mathfrak{K}$ a family of groups, the *homology groups*. The $d^{\text{th}}$ homology group $\mathrm{H}_d(\mathfrak{K})$ of $\mathfrak{K}$ contains $d$-dimensional topological features, such as connected components ($d = 0$), cycles/tunnels ($d = 1$), and voids ($d = 2$). Homology groups are typically summarised by their ranks, thereby obtaining a simple invariant "signature" of a manifold. For example, a circle in $\mathbb{R}^2$ has one feature with $d = 1$ (a cycle), and one feature with $d = 0$ (a connected component). In practice, the underlying manifold $\mathbb{M}$ is unknown and we are working with a point cloud $X := \{x_1, \dots, x_n\} \subseteq \mathbb{R}^d$ and a metric $\mathrm{dist}\colon X \times X \to \mathbb{R}$ such as the Euclidean distance. Persistent homology extends simplicial homology to this setting: instead of approximating $\mathbb{M}$ by means of a *single* simplicial complex, which would

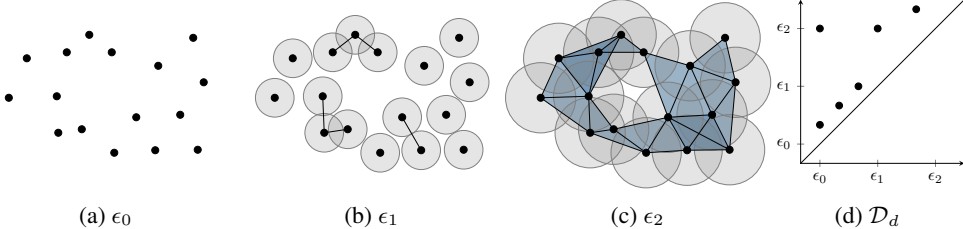

(a) $\epsilon_0$      (b) $\epsilon_1$      (c) $\epsilon_2$      (d) $\mathcal{D}_d$

Figure 1: The Vietoris–Rips complex $\mathfrak{R}_\epsilon(X)$ of a point cloud $X$ at different scales $\epsilon_0$, $\epsilon_1$, and $\epsilon_2$. As the distance threshold $\epsilon$ increases, the connectivity changes. The creation and destruction of $d$-dimensional topological features is recorded in the $d^{\text{th}}$ persistence diagram $\mathcal{D}_d$.

be an unstable procedure due to the discrete nature of $X$, persistent homology tracks changes in the homology groups over *multiple* scales of the metric. This is achieved by constructing a special simplicial complex, the Vietoris–Rips complex (Vietoris, 1927). For $0 \leq \epsilon < \infty$, the Vietoris–Rips complex of $X$ at scale $\epsilon$, denoted by $\mathfrak{R}_\epsilon(X)$, contains all simplices (i.e. subsets) of $X$ whose elements $\{x_0, x_1, \dots\}$ satisfy $\text{dist}(x_i, x_j) \leq \epsilon$ for all $i, j$. Given a matrix $\mathbf{A}$ of pairwise distances of a point cloud $X$, we will use $\mathfrak{R}_\epsilon(\mathbf{A})$ and $\mathfrak{R}_\epsilon(X)$ interchangeably because constructing $\mathfrak{R}_\epsilon$ only requires distances. Vietoris–Rips complexes satisfy a nesting relation, i.e. $\mathfrak{R}_{\epsilon_i}(X) \subseteq \mathfrak{R}_{\epsilon_j}(X)$ for $\epsilon_i \leq \epsilon_j$, making it possible to track changes in the homology groups as $\epsilon$ increases (Edelsbrunner et al., 2002). Figure 1 illustrates this process. Since $X$ contains a finite number of points, a maximum $\tilde{\epsilon}$ value exists for which the connectivity stabilises; therefore, calculating $\mathfrak{R}_{\tilde{\epsilon}}$ is sufficient to obtain topological features at all scales.

We write $\text{PH}(\mathfrak{R}_\epsilon(X))$ for the persistent homology calculation of the Vietoris–Rips complex. It results in a tuple $(\{\mathcal{D}_1, \mathcal{D}_2, \dots\}, \{\pi_1, \pi_2, \dots\})$ of *persistence diagrams* (1st component) and *persistence pairings* (2nd component). The $d$-dimensional persistence diagram $\mathcal{D}_d$ (Figure 1d) of $\mathfrak{R}_\epsilon(X)$ contains coordinates of the form $(a, b)$, where $a$ refers to a threshold $\epsilon$ at which a $d$-dimensional topological feature is created in the Vietoris–Rips complex, and $b$ refers to a threshold $\epsilon'$ at which it is destroyed. When $d = 0$, for example, the threshold $\epsilon'$ indicates at which distance two connected components in $X$ are merged into one; this is related to spanning trees (Kurlin, 2015). The $d$-dimensional persistence pairing contains indices $(i, j)$ corresponding to simplices $s_i, s_j \in \mathfrak{R}_\epsilon(X)$ that create and destroy the topological feature identified by $(a, b) \in \mathcal{D}_d$, respectively. Persistence diagrams are known to be stable with respect to small perturbations in the data set (Cohen-Steiner et al., 2007). Two diagrams $\mathcal{D}$ and $\mathcal{D}'$ can be compared using the *bottleneck distance* $\text{d}_\text{b}(\mathcal{D}, \mathcal{D}') := \inf_{\eta:\ \mathcal{D} \to \mathcal{D}'} \sup_{x \in \mathcal{D}} \|x - \eta(x)\|_\infty$, where $\eta \colon \mathcal{D} \to \mathcal{D}'$ denotes a bijection between the points of the two diagrams, and $\| \cdot \|_\infty$ refers to the $\text{L}_\infty$ norm. We use $\mathcal{D}^X$ to refer to the set of persistence diagrams of a point cloud $X$ arising from $\text{PH}(\mathfrak{R}_\epsilon(X))$.

## 3 PROPOSED METHOD

We propose a generic framework for constraining autoencoders to preserve topological structures (measured via persistent homology) of the data space in their latent encodings.

### 3.1 VIETORIS–RIPS COMPLEX CALCULATION

Given a finite metric space $\mathcal{S}$, we first calculate the persistent homology of the Vietoris–Rips complex of its distance matrix $\mathbf{A}^\mathcal{S}$. Typically, the usual Euclidean distance is used to calculate $\mathbf{A}^\mathcal{S}$, but both the persistent homology calculation and our method are *not* restricted to any particular distance; previous research (Wagner & Dłotko, 2014) shows that even similarity measures that do not satisfy the properties of a metric can be used successfully with $\text{PH}(\cdot)$. Subsequently, let $\epsilon := \max \mathbf{A}^\mathcal{S}$ so that $\mathfrak{R}_\epsilon(\mathbf{A}^\mathcal{S})$ is the corresponding Vietoris–Rips complex as described in Section 2. Given a maximum dimension[1] of $d \in \mathbb{N}_{>0}$, we obtain a set of persistence diagrams $\mathcal{D}^\mathcal{S}$, and a set of persistence pairings $\pi^\mathcal{S}$. The $d^{\text{th}}$ persistence pairing $\pi_d^\mathcal{S}$ contains indices of simplices that are relevant for creating and

---

[1]This means that we do not have to consider higher-dimensional topological features, making the calculation more efficient.

destroying $d$-dimensional topological features. We can consider each pairing to represent *edge indices*, namely the edges that are deemed to be "topologically relevant" by the computation of persistent homology (see below for more details). This works because the Vietoris–Rips complex is a *clique complex*, i.e. it is fully determined by its edges (Zomorodian, 2010).

For $0$-dimensional topological features, it is sufficient to look at the *edge indices*, i.e. the indices of "destroyer" simplices, contained in $\pi_0^{\mathcal{S}}$. Each of these indices corresponds to an edge in the minimum spanning tree of the data set. This calculation is computationally efficient, having a worst-case complexity of $\mathcal{O}\big(m^2 \cdot \alpha\big(m^2\big)\big)$, where $m$ is the batch size and $\alpha(\cdot)$ denotes the extremely slow-growing inverse Ackermann function (Cormen et al., 2009, Chapter 22). For $1$-dimensional features, where edges are paired with triangles, we obtain edge indices by selecting the edge with the maximum weight of the triangle. While this procedure (and thus our method) generalises to higher dimensions, our current implementation supports no higher-dimensional features. Since preliminary experiments showed that including $1$-dimensional topological features merely increases runtime, we will focus only on $0$-dimensional persistence diagrams in the subsequent experiments. We thus use $\big(\mathcal{D}^{\mathcal{S}}, \pi^{\mathcal{S}}\big)$ to denote *the* $0$-dimensional persistence diagram and pairing of $\mathcal{S}$, respectively.

## 3.2 TOPOLOGICAL AUTOENCODER

In the following, we consider a mini-batch $X$ of size $m$ from the data space $\mathcal{X}$ as a point cloud. Furthermore, we define an autoencoder as the composition of two functions $h \circ g$, where $g \colon \mathcal{X} \to \mathcal{Z}$ represents the *encoder* and $h \colon \mathcal{Z} \to \mathcal{X}$ represents the *decoder*. We denote latent codes with $Z := g(X)$. During a forward pass of the autoencoder, we compute the persistent homology of the mini-batch in both the data as well as the latent space, yielding two sets of tuples, i.e. $\big(\mathcal{D}^X, \pi^X\big) := \mathrm{PH}(\mathfrak{R}_\epsilon(X))$ and $\big(\mathcal{D}^Z, \pi^Z\big) := \mathrm{PH}(\mathfrak{R}_\epsilon(Z))$. The values of the persistence diagram can be retrieved by subsetting the distance matrix with the indices provided by the persistence pairings; we write $\mathcal{D}^X \simeq \mathbf{A}^X\big[\pi^X\big]$ to indicate that the diagram, which is a set, contains the same information as the distances we retrieve with the pairing. We treat $\mathbf{A}^X\big[\pi^X\big]$ as a vector in $\mathbb{R}^{|\pi^X|}$. Informally speaking, the persistent homology calculation can thus be seen as a selection of topologically relevant edges of the Vietoris–Rips complex, followed by the selection of corresponding entries in the distance matrix. By comparing both diagrams, we can construct a topological regularisation term $\mathcal{L}_t := \mathcal{L}_t\big(\mathbf{A}^X, \mathbf{A}^Z, \pi^X, \pi^Z\big)$, which we add to the reconstruction loss of an autoencoder, i.e.

$$\mathcal{L} := \mathcal{L}_r(X, h(g(X))) + \lambda \, \mathcal{L}_t \tag{1}$$

where $\lambda \in \mathbb{R}$ is a parameter to control the strength of the regularisation. Next, we discuss how to specify $\mathcal{L}_t$: in our case, the PH calculation represents a selection of topologically relevant distances from the distance matrix. Each persistence diagram entry corresponds to a distance between two data points. We assume that the distances are *unique* so that each entry in the diagram has an infinitesimal neighbourhood that only contains a single point. Given this fixed pairing and a differentiable distance function, the persistence diagram entries are thus also differentiable with respect to the encoder parameter. Hence, the persistence pairing does not change upon a small perturbation of the underlying distances, thereby guaranteeing the existence of the derivative of our loss function. This, in turn, permits the calculation of gradients for backpropagation.

A straightforward approach to impose the data space topology on the latent space would be to directly calculate a loss based on the selected distances in both spaces. Such an approach will *not* result in informative gradients for the autoencoder, as it merely compares topological features without matching[2] the edges between $\mathfrak{R}_\epsilon(X)$ and $\mathfrak{R}_\epsilon(Z)$. A cleaner approach would be to enforce similarity on the intersection of the selected edges in both complexes. However, this would initially include very few edges, preventing efficient training and leading to very biased estimates of the topological alignments between the spaces[3]. To overcome this, we account for the *union* of all selected edges in $X$ and $Z$. Our topological loss term decomposes into two components, each

---

[2]We use the term "matching" only to build intuition. Our approach does not calculate a matching in the sense of a bottleneck or Wasserstein distance between persistence diagrams.

[3]When initialising a random latent space $Z$, the persistence pairing in the latent space will select random edges, resulting in only 1 expected matched edge (independent of mini-batch size) between the two pairings. Thus, only one edge (referring to one pairwise distance between two latent codes) could be used to update the encoding of these two data points.

handling the "directed" loss occurring as topological features in one of the two spaces remain fixed. Hence, $\mathcal{L}_t = \mathcal{L}_{\mathcal{X} \to \mathcal{Z}} + \mathcal{L}_{\mathcal{Z} \to \mathcal{X}}$, where

$$\mathcal{L}_{\mathcal{X} \to \mathcal{Z}} := \frac{1}{2} \left\| \mathbf{A}^X \left[ \pi^X \right] - \mathbf{A}^Z \left[ \pi^X \right] \right\|^2 \quad \text{and} \quad \mathcal{L}_{\mathcal{Z} \to \mathcal{X}} := \frac{1}{2} \left\| \mathbf{A}^Z \left[ \pi^Z \right] - \mathbf{A}^X \left[ \pi^Z \right] \right\|^2, \quad (2)$$

respectively. The key idea for both terms is to align and preserve topologically relevant distances from both spaces. By taking the union of all selected edges (and the corresponding distances), we obtain an informative loss term that is determined by at least $|X|$ distances. This loss can be seen as a more generic version of the loss introduced by Hofer et al. (2019b), whose formulation does not take the two directed components into account and optimises the destruction values of all persistence tuples with respect to a uniform parameter (see also Section 4 for a brief discussion). By contrast, our formulation aims to to align the distances between $X$ and $Z$ (which in turn will lead to an alignment of distances between $\mathcal{X}$ and $\mathcal{Z}$). If they are aligned perfectly, $\mathcal{L}_{\mathcal{X} \to \mathcal{Z}} = \mathcal{L}_{\mathcal{Z} \to \mathcal{X}} = 0$ because both pairings and their corresponding distances coincide. The converse implication is not true: if $\mathcal{L}_t = 0$, the persistence pairings (and their corresponding persistence diagrams) are not necessarily identical.

Letting $\boldsymbol{\theta}$ refer to the parameters of the *encoder*, we have

$$\frac{\partial}{\partial \boldsymbol{\theta}} \mathcal{L}_{\mathcal{X} \to \mathcal{Z}} = \frac{\partial}{\partial \boldsymbol{\theta}} \left( \frac{1}{2} \left\| \mathbf{A}^X \left[ \pi^X \right] - \mathbf{A}^Z \left[ \pi^X \right] \right\|^2 \right) = - \left( \mathbf{A}^X \left[ \pi^X \right] - \mathbf{A}^Z \left[ \pi^X \right] \right)^\top \left( \frac{\partial \mathbf{A}^Z \left[ \pi^X \right]}{\partial \boldsymbol{\theta}} \right) \quad (3)$$

$$= - \left( \mathbf{A}^X \left[ \pi^X \right] - \mathbf{A}^Z \left[ \pi^X \right] \right)^\top \left( \sum_{i=1}^{|\pi^X|} \frac{\partial \mathbf{A}^Z \left[ \pi^X \right]_i}{\partial \boldsymbol{\theta}} \right), \quad (4)$$

where $\left| \pi^X \right|$ denotes the cardinality of a persistence pairing and $\mathbf{A}^Z \left[ \pi^X \right]_i$ refers to the $i^{\text{th}}$ entry of the vector of paired distances. This derivation works analogously for $\mathcal{L}_{\mathcal{Z} \to \mathcal{X}}$ (with $\pi^X$ being replaced by $\pi^Z$). Furthermore, any derivative of $\mathbf{A}^X$ with respect to $\boldsymbol{\theta}$ must vanish because the distances of the input samples do not depend on the encoding by definition.

These equations presume infinitesimal perturbations. In fact, the persistence diagrams change in a non-differentiable manner during the training phase but for a given update step, a diagram is robust to infinitesimal changes of its entries (Cohen-Steiner et al., 2007). As a consequence, our topological loss is differentiable for each update step during training. We make our code publicly available[4].

## 3.3 STABILITY

With persistence diagrams being stable under small perturbations of the underlying space (Cohen-Steiner et al., 2007), we still have to analyse our topological approximation on the level of mini-batches. The following theorem guarantees that subsampled persistence diagrams are close to the persistence diagrams of the original point cloud.

**Theorem 1.** *Let $X$ be a point cloud of cardinality $n$ and $X^{(m)}$ be one subsample of $X$ of cardinality $m$, i.e. $X^{(m)} \subseteq X$, sampled without replacement. We can bound the probability of the persistence diagrams of $X^{(m)}$ exceeding a threshold in terms of the bottleneck distance as*

$$\mathbb{P} \left( \mathrm{d}_\mathrm{b} \left( \mathcal{D}^X, \mathcal{D}^{X^{(m)}} \right) > \epsilon \right) \leq \mathbb{P} \left( \mathrm{d}_\mathrm{H} \left( X, X^{(m)} \right) > 2\epsilon \right), \quad (5)$$

*where $\mathrm{d}_\mathrm{H}$ refers to the Hausdorff distance between the point cloud and its subsample.*

*Proof.* See Section A.1 in the supplementary materials. $\square$

For $m \to n$, we have $\lim_{m \to n} \mathrm{d}_\mathrm{H} \left( X, X^{(m)} \right) = 0$. Please refer to Section A.2 for an analysis of empirical convergence rates as well as a discussion of a worst-case bound. Given certain independence assumptions, the next theorem approximates the expected value of the Hausdorff distance. The calculation of an exact representation is beyond the scope of this work, though.

---

[4]https://osf.io/abuce/?view_only=f16d65d3f73e4918ad07cdd08a1a0d4b

**Theorem 2.** *Let* $\mathbf{A} \in \mathbb{R}^{n \times m}$ *be the distance matrix between samples of* $X$ *and* $X^{(m)}$*, where the rows are sorted such that the first* $m$ *rows correspond to the columns of the* $m$ *subsampled points with diagonal elements* $a_{ii} = 0$*. Assume that the entries* $a_{ij}$ *with* $i > m$ *are random samples following a distance distribution* $F_D$ *with* $\operatorname{supp}(f_D) \in \mathbb{R}_{\geq 0}$*. The minimal distances* $\delta_i$ *for rows with* $i > m$ *follow a distribution* $F_\Delta$*. Letting* $Z := \max_{1 \leq i \leq n} \delta_i$ *with a corresponding distribution* $F_Z$*, the expected Hausdorff distance between* $X$ *and* $X^{(m)}$ *for* $m < n$ *is bounded by:*

$$\mathbb{E}\left[\mathrm{d_H}(X, X^{(m)})\right] = \mathbb{E}_{Z \sim F_Z}[Z] \leq \int_0^{+\infty} \left(1 - F_D(z)^{(n-1)}\right) \mathrm{d}z \leq \int_0^{+\infty} \left(1 - F_D(z)^{m(n-m)}\right) \mathrm{d}z \quad (6)$$

*Proof.* See Section A.3 in the supplementary materials. □

From Eq. 6, we obtain $\mathbb{E}[\mathrm{d_H}(X, X^m)] = 0$ as $m \to n$, so the expected value converges as the subsample size approaches the total sample size[5]. We conclude that our subsampling approach results in point clouds that are suitable proxies for the large-scale topological structures of the point cloud $X$.

## 4 RELATED WORK

Computational topology and persistent homology (PH) have started gaining traction in several areas of machine learning research. PH is often used as as *post hoc* method for analysing topological characteristics of data sets. Thus, there are several methods that compare topological features of high-dimensional spaces with different embeddings to assess the fidelity and quality of a specific embedding scheme (Khrulkov & Oseledets, 2018; Paul & Chalup, 2017; Rieck & Leitte, 2015; 2017; Yan et al., 2018). PH can also be used to characterise the training of neural networks (Guss & Salakhutdinov, 2018; Rieck et al., 2019b), as well as their decision boundaries (Ramamurthy et al., 2019). Our method differs from all these publications in that we are able to obtain gradient information to *update* a model while training. Alternatively, topological features can be integrated into classifiers to improve classification performance. Hofer et al. (2017) propose a neural network layer that learns projections of persistence diagrams, which can subsequently be used as feature descriptors to classify structured data. Moreover, several vectorisation strategies for persistence diagrams exist (Adams et al., 2017; Carrière et al., 2015), making it possible to use them in kernel-based classifiers. These strategies have been subsumed (Carrière et al., 2019) in a novel architecture based on deep sets. The commonality of these approaches is that they treat persistence diagrams as being fixed; while they are capable of learning suitable parameters for classifying them, they cannot adjust input data to better approximate a certain topology. Such topology-based adjustments have only recently become feasible. Poulenard et al. (2018) demonstrated how to optimise real-valued functions based on their topology. This constitutes the first approach for aligning persistence diagrams by modifying input data; it requires the connectivity of the data to be known, and the optimised functions have to be node-based and scalar-valued. By contrast, our method works *directly* on distances and sidesteps connectivity calculations via the Vietoris–Rips complex. Chen et al. (2019) use a similar optimisation technique to regularise the decision boundary of a classifier. However, this requires discretising the space, which can be computationally expensive. Hofer et al. (2019b), the closest work to ours, also presents a differentiable loss term. They propose regularising the connectivity of the latent space by itself. Specifically, their formulation enforces a single scale on the latent space. The learned encoding is then applied to a classification task. By contrast, our directed loss term aims to *preserve* the data space topology in the latent space for dimensionality reduction.

## 5 EXPERIMENTS

Our main task is to learn a latent space in an unsupervised manner such that topological features of the data space—measured using persistent homology approximations on every batch—are preserved as much as possible.

---

[5]For $m = n$, the two integrals switch their order as $m(n-m) = 0 < n-1$ (for $n > 1$).

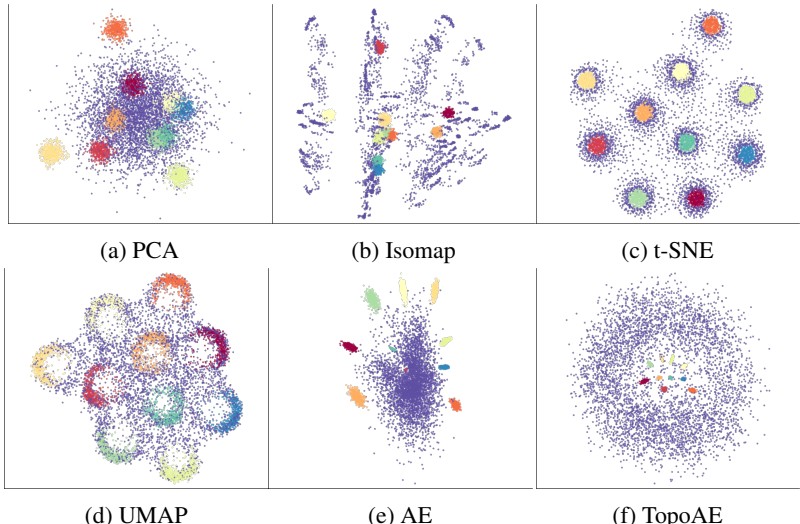

|  |  |  |
|:--:|:--:|:--:|
| (a) PCA | (b) Isomap | (c) t-SNE |
| (d) UMAP | (e) AE | (f) TopoAE |

Figure 2: Latent representations of the SPHERES data set. Only our method is capable of representing the complicated nesting relationship inherent to the data. TopoAE used a batch-size of 28. See Figure A.2 in the supplementary materials for an enlarged version.

## 5.1 EXPERIMENTAL SETUP

**Data sets** We use a SPHERES data set that consists of ten high-dimensional 100-spheres living in $101-$dimensional space that are enclosed by one larger sphere that consists of the same number of points as the total of inner spheres (please refer to Section A.4 for more details). We also use three image data sets (MNIST, FASHION-MNIST, and CIFAR-10), which are particularly amenable to our topology-based analysis because real-world images are known to lie *on* or *near* low-dimensional manifolds (Lee et al., 2003; Peyré, 2009).

**Baselines & Training** We compare our approach with several dimensionality reduction techniques, including UMAP (McInnes et al., 2018), t-SNE (van der Maaten & Hinton, 2008), Isomap (Tenenbaum et al., 2000), PCA, as well as standard autoencoders (AE). We apply our proposed topological constraint to this standard autoencoder architecture (TopoAE). For comparability and interpretability, each method is restricted to two latent dimensions. We split each data set into training and testing (using the predefined split if available; 90% versus 10% otherwise). Additionally, we remove 15% of the training split as a validation data set for tuning the hyperparameters. We normalised our topological loss term by the batch size $m$ in order to disentangle $\lambda$ from it. All autoencoders employ batch-norm and are optimized using ADAM (Kingma & Ba, 2014). Since t-SNE is not intended to be applied applying to previously unseen test samples, we evaluate this method only on the train split. In addition, significant computational scaling issues forced us to forgo running a hyperparameter search for Isomap on real-world data sets, so we only compare this algorithm on the synthetic data set. Please refer to Section A.5 for more details on architectures and hyperparameters.

**Evaluation** We evaluate the quality of latent representations in terms of (1) low-dimensional visualisations, (2) dimensionality reduction quality metrics (evaluated between input data and *latent* codes), and (3) reconstruction errors (Data MSE; evaluated between input and *reconstructed* data), provided that invertible transformations are available[6]. For (2), we consider several measures (please refer to Section A.6 for more details). First, we calculate $\mathrm{KL}_\sigma$, the Kullback–Leibler divergence between the density estimates of the input and latent space, based on density estimates (Chazal et al., 2011; 2014b), where $\sigma \in \mathbb{R}_{>0}$ denotes the length scale of the Gaussian kernel, which is varied to account for multiple data scales. We chose minimising $\mathrm{KL}_{0.1}$ as our hyperparameter search objective. Furthermore, we calculate common non-linear dimensionality reduction (NLDR) quality metrics, which use the pairwise distance matrices of the input and the *latent* space (as indicated by the "$\ell$" in the abbreviations), namely the *root mean square error* ($\ell$-RMSE), which—despite its name—is not

---

[6]Invertible transformations are available for PCA and all autoencoder-based methods.

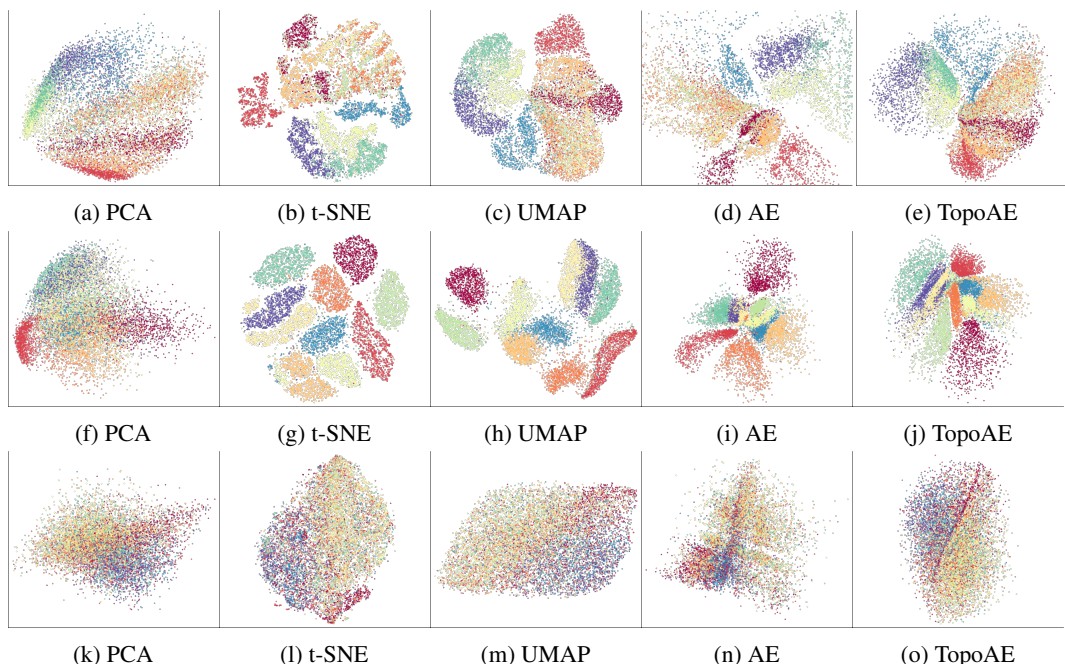

|  |  |  |  |  |
|---|---|---|---|---|
| (a) PCA | (b) t-SNE | (c) UMAP | (d) AE | (e) TopoAE |
| (f) PCA | (g) t-SNE | (h) UMAP | (i) AE | (j) TopoAE |
| (k) PCA | (l) t-SNE | (m) UMAP | (n) AE | (o) TopoAE |

Figure 3: Latent representations of the FASHION-MNIST (top row), MNIST (middle row), CIFAR-10 (bottom row) data sets. TopoAE batch-sizes (same data set order): $(95, 126, 82)$. Please refer to Figures A.3, A.4, and A.5 in the supplementary materials for enlarged versions.

related to the reconstruction error of the autoencoder but merely measures to what extent the two distributions of distances coincide, the *mean relative rank error* ($\ell$-MRRE), the *continuity* ($\ell$-Cont), and the *trustworthiness* ($\ell$-Trust) . The reported measures are computed on the testing splits (except for t-SNE where no transformation between splits is available, so we report the measures on a random subsample of the train split).

## 5.2  RESULTS

**Quantitative Results** Table 1 reports the quantitative results. Overall, we observe that our method shines in preserving the data density over multiple length scales (as measured by KL). Furthermore, we find that TopoAE displays competitive continuity values ($\ell$-Cont) and reconstruction errors (Data MSE). The remaining classical measures favour the baselines (foremost the *train* (!) performance of t-SNE). However, we will subsequently see that those classic measures *fail* at detecting the relevant structural information, as exemplified with known ground truth manifolds, such as the SPHERES data set.

**Visualisation of latent spaces** For the SPHERES data set (Figure 2), we observe that only our method is capable of assessing the nesting relationship of the high-dimensional spheres correctly. By contrast, t-SNE "cuts open" the enclosing sphere, distributing most of its points around the remaining spheres. We see that the KL-divergence confirms the visual assessment that only our proposed method preserves the relevant structure of this dataset. Several classical evaluation measures, however, favour t-SNE, even though this method fails to capture the global structure and nesting relationship of the enclosing sphere manifold accounting for half of the dataset. On FASHION-MNIST (Figure 3a–e), we see that, as opposed to AE, which is purely driven by the reconstruction error, our method has the additional objective of *preserving* structure. Here, this constraint helps the autoencoder to "organise" the latent space, resulting in a comparable pattern as in UMAP, which is also topologically motivated. Furthermore, we observe that t-SNE tends to fragment certain classes (dark orange, red) into multiple distinct subgroups. This is very likely not to reflect the underlying manifold structure, but constitutes an artefact frequently observed with this method. For MNIST, the latent embeddings (Figure 3f–j) demonstrate that the non-linear competitors—mostly by pulling apart distinct classes—lose some

Table 1: Multiple evaluation metrics (Section 5.1). The hyperparameters of all tunable methods were selected to minimise the objective $KL_{0.1}$. The winner is shown in bold and underlined, the runner-up in bold. Please refer to Supplementary Table A.2 for more $\sigma$ scales and variance estimates.

| Data set | Method | $KL_{0.01}$ | $KL_{0.1}$ | $KL_1$ | $\ell$-MRRE | $\ell$-Cont | $\ell$-Trust | $\ell$-RMSE | Data MSE |
|---|---|---|---|---|---|---|---|---|---|
| SPHERES | Isomap | 0.181 | **0.420** | **0.00881** | **0.246** | **0.790** | **0.676** | 10.4 | – |
| | PCA | 0.332 | 0.651 | 0.01530 | 0.294 | 0.747 | 0.626 | 11.8 | 0.9610 |
| | TSNE | **0.152** | 0.527 | 0.01271 | 0.217 | 0.773 | 0.679 | 8.1 | – |
| | UMAP | 0.157 | 0.613 | 0.01658 | 0.250 | 0.752 | 0.635 | **9.3** | – |
| | AE | 0.566 | 0.746 | 0.01664 | 0.349 | 0.607 | 0.588 | 13.3 | 0.8155 |
| | TopoAE | 0.085 | 0.326 | 0.00694 | 0.272 | 0.822 | 0.658 | 13.5 | **0.8681** |
| F-MNIST | PCA | 0.356 | 0.052 | 0.00069 | 0.057 | 0.968 | 0.917 | 9.1 | 0.1844 |
| | TSNE | 0.405 | 0.071 | 0.00198 | 0.020 | 0.967 | **0.974** | 41.3 | – |
| | UMAP | 0.424 | 0.065 | 0.00163 | 0.029 | 0.981 | 0.959 | **13.7** | – |
| | AE | 0.478 | 0.068 | 0.00125 | **0.026** | 0.968 | **0.974** | 20.7 | 0.1020 |
| | TopoAE | **0.392** | **0.054** | **0.00100** | 0.032 | **0.980** | 0.956 | 20.5 | **0.1207** |
| MNIST | PCA | 0.389 | 0.163 | 0.00160 | 0.166 | 0.901 | 0.745 | 13.2 | 0.2227 |
| | TSNE | 0.277 | **0.133** | 0.00214 | 0.040 | 0.921 | 0.946 | 22.9 | – |
| | UMAP | **0.321** | 0.146 | 0.00234 | **0.051** | 0.940 | 0.938 | **14.6** | – |
| | AE | 0.620 | 0.155 | **0.00156** | 0.058 | 0.913 | 0.937 | 18.2 | 0.1373 |
| | TopoAE | 0.341 | 0.110 | 0.00114 | 0.056 | **0.932** | 0.928 | 19.6 | **0.1388** |
| CIFAR | PCA | **0.591** | 0.020 | 0.00023 | 0.119 | 0.931 | 0.821 | 17.7 | 0.1482 |
| | TSNE | 0.627 | 0.030 | 0.00073 | 0.103 | 0.903 | **0.863** | **25.6** | – |
| | UMAP | 0.617 | 0.026 | 0.00050 | 0.127 | 0.920 | 0.817 | 33.6 | – |
| | AE | 0.668 | 0.035 | 0.00062 | 0.132 | 0.851 | 0.864 | 36.3 | **0.1403** |
| | TopoAE | 0.556 | 0.019 | 0.00031 | **0.108** | 0.927 | 0.845 | 37.9 | 0.1398 |

of the relationship information *between* clusters when comparing against our proposed method or PCA. Finally, we observe that CIFAR-10 (Figure 3k–o), is challenging to embed in two latent dimensions in a purely unsupervised manner. Interestingly, our method (repeatedly) identified a linear substructure separating the latent space in two additional groups of classes.

## 6 DISCUSSION AND CONCLUSIONS

We presented topological autoencoders, a novel method for preserving topological information, measured in terms of persistent homology, of the input space when learning latent representations with deep neural networks. Under weak theoretical assumptions, we showed how our persistent homology (PH) calculations can be combined with backpropagation; moreover, we proved that approximating PH on the level of mini-batches is theoretically justified.

In our experiments, we observed that our method is uniquely able to capture spatial relationships between nested high-dimensional spheres. This is relevant, as the ability to cope with *several* manifolds in the domain of manifold learning still remains a challenging task. On real-world data sets, we observed that our topological loss leads to competitive performance in terms of numerous quality metrics (such as a density preservation metric), while not adversely affecting the reconstruction error. In both synthetic and real-world data sets, we obtain interesting representations, as our method does not merely pull apart different classes, but tries to spatially arrange them meaningfully. Thus, we do not observe mere distinct "clouds", but rather entangled structures, which we consider to constitute a more meaningful representation of the underlying manifolds (an auxiliary analysis in Supplementary Section A.9 confirms that our method influences topological features, measured using PH, in a beneficial manner).

Our topological loss formulation is highly generic; it only requires the existence of a distance matrix between individual samples (either globally, or on the level of batches). As a consequence, our topological loss term can be directly integrated into a variety of different architectures and is *not*

limited to standard autoencoders. For instance, we can also apply our constraint to variational setups (see Figure A.7 for a sketch).

Employing our generic constraint to more involved architectures will be an exciting route for future work. One issue with the calculation is that, given the computational complexity of calculating $\mathfrak{R}_\epsilon(\cdot)$, for higher-dimensional features we would scale progressively worse with increasing batch size. However, in our low-dimensional setup, we observed that runtime tends to grow with decreasing batch-size, i.e. the mini-batch speed-up still dominates runtime (for more details concerning the effect of batch sizes, see Supplementary Section A.7). In future work, scaling to higher dimensions could be mitigated by approximating the calculation of persistent homology (Choudhary et al., 2018; Kerber & Sharathkumar, 2013; Sheehy, 2013) or by exploiting recent advances in parallelising it (Bauer et al., 2014; Lewis & Morozov, 2015).

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

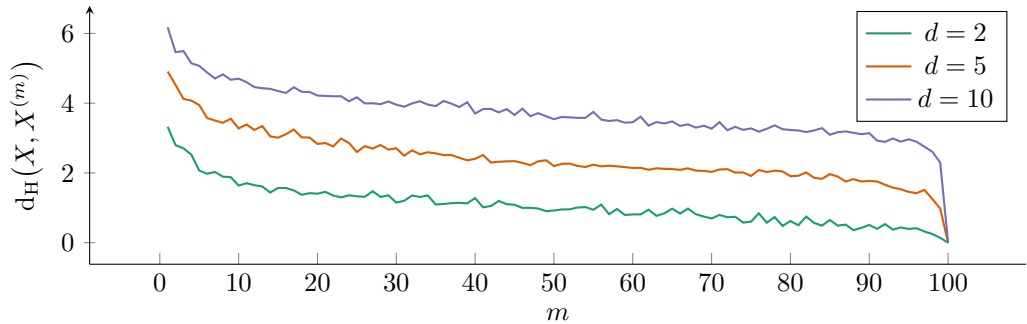

Figure A.1: Empirical convergence rate (mean) of the Hausdorff distance for a subsample of size $m$ of 100 points in a $d$-dimensional space, following a standard normal distribution.

## A   APPENDIX

### A.1   PROOF OF THEOREM 1

**Theorem 1.** *Let $X$ be a point cloud of cardinality $n$ and $X^{(m)}$ be one subsample of $X$ of cardinality $m$, i.e. $X^{(m)} \subseteq X$, sampled without replacement. We can bound the probability of $X^{(m)}$ exceeding a threshold in terms of the bottleneck distance as*

$$\mathbb{P}\left(\mathrm{d_b}\left(\mathcal{D}^X, \mathcal{D}^{X^{(m)}}\right) > \epsilon\right) \leq \mathbb{P}\left(\mathrm{d_H}\left(X, X^{(m)}\right) > 2\epsilon\right), \tag{7}$$

*where $\mathrm{d_H}$ refers to the Hausdorff distance between the point cloud and its subsample, i.e.*

$$\mathrm{d_H}(X, Y) := \max\left\{\sup_{x \in X} \inf_{y \in Y} \mathrm{dist}(x, y), \sup_{y \in Y} \inf_{x \in X} \mathrm{dist}(x, y)\right\} \tag{8}$$

*for a baseline distance $\mathrm{dist}(x, y)$ such as the Euclidean distance.*

*Proof.* The stability of persistent homology calculations was proved by Chazal et al. (2014a) for finite metric spaces. More precisely, given two metric spaces $X$ and $Y$, we have

$$\mathrm{d_b}\left(\mathcal{D}^X, \mathcal{D}^Y\right) \leq 2\,\mathrm{d_{GH}}(X, Y), \tag{9}$$

where $\mathrm{d_{GH}}(X, Y)$ refers to the Gromov–Hausdorff distance (Burago et al., 2001, p. 254) of the two spaces. It is defined as the infimum Hausdorff distance over all isometric embeddings of $X$ and $Y$. This distance can be employed for shape comparison (Chazal et al., 2009; Mémoli & Sapiro, 2004), but is hard to compute. In our case, with $X = X$ and $Y = X^{(m)}$, we consider both spaces to have the same metric (for $Y$, we take the canonical restriction of the metric from $X$ to the subspace $Y$). By definition of the Gromov–Hausdorff distance, we thus have $\mathrm{d_{GH}}(X, Y) \leq \mathrm{d_H}(X, Y)$, so Eq. 8 leads to

$$\mathrm{d_b}\left(\mathcal{D}^X, \mathcal{D}^Y\right) \leq 2\,\mathrm{d_H}(X, Y), \tag{10}$$

from which the original claim from Eq. 7 follows by taking probabilities on both sides. $\square$

### A.2   EMPIRICAL CONVERGENCE RATES OF $\mathrm{d_H}\left(X, X^{(m)}\right)$

Figure A.1 depicts the mean of the convergence rate (mean) of the Hausdorff distance for a sub-sample of size $m$ of 100 points in a $d$-dimensional space, following a standard normal distribution. We can see that the convergence rate is roughly similar, but shown on different absolute levels that depend on the ambient dimension. While bounding the convergence rate of this expression is feasible (Chazal et al., 2015a;b), it requires more involved assumptions on the measures from which $X$ and $X^{(m)}$ are sampled. Additionally, we can give a simple bound using the *diameter* $\mathrm{diam}(X) := \sup\{\mathrm{dist}(x, y) \mid x, y \in X\}$. We have $\mathrm{d_H}\left(X, X^{(m)}\right) \leq \mathrm{diam}(X)$ because the supremum is guaranteed to be an upper bound for the Hausdorff distance. This worst-case bound does not account for the sample size (or mini-batch size) $m$, though (see Theorem 2 for an expression that takes $m$ into account).

### A.3 PROOF OF THEOREM 2

Prior to the proof we state two observations that arise from our special setting of dealing with finite point clouds.

**Observation 1.** *Since $X^{(m)} \subseteq X$, we have $\sup_{x' \in X^{(m)}} \inf_{x \in X} \mathrm{dist}(x, x') = 0$. Hence, the Hausdorff distance simplifies to:*

$$\mathrm{d_H}\left(X, X^{(m)}\right) := \sup_{x \in X} \inf_{x' \in X^m} \mathrm{dist}(x, x') \tag{11}$$

*In other words, we only have to consider a "one-sided" expression of the distance because the distance from the subsample to the original point cloud is always zero.*

**Observation 2.** *Since our point clouds of interest are finite sets, the suprema and infima of the Hausdorff distance coincide with the maxima and minima, which we will subsequently use for easier readability.*

Hence, the computation of $\mathrm{d_H}(X, X^{(m)})$ can be divided into three steps.

1. Using the baseline distance $\mathrm{dist}(\cdot, \cdot)$, we compute a distance matrix $\mathbf{A} \in \mathbb{R}^{n \times m}$ between all points in $X$ and $X^{(m)}$.

2. For each of the $n$ points in $X$, we compute the minimal distance to the $m$ samples of $X^{(m)}$ by extracting the minimal distance per row of $\mathbf{A}$ and gather all minimal distances in $\boldsymbol{\delta} \in \mathbb{R}^n$.

3. Finally, we return the maximal entry of $\boldsymbol{\delta}$ as $\mathrm{d_H}\left(X, X^{(m)}\right)$.

In the subsequent proof, we require an independence assumption of the samples.

*Proof.* Using Observations 1 and 2 we obtain a simplified expression for the Hausdorff distance, i.e.

$$\mathrm{d_H}\left(X, X^{(m)}\right) := \max_{i, 1 \leq i \leq n} \left( \min_{j, 1 \leq j \leq m} (a_{ij}) \right). \tag{12}$$

The minimal distances of the first $m$ rows of $\mathbf{A}$ are trivially 0. Hence, the outer maximum is determined by the remaining $n - m$ row minima $\{\delta_i \mid m < i \leq n\}$ with $\delta_i = \min_{1 \leq j \leq m} (a_{ij})$. Those minima follow the distribution $F_\Delta(y)$ with

$$F_\Delta(y) = \mathbb{P}(\delta_i \leq y) = 1 - \mathbb{P}(\delta_i > y) = 1 - \mathbb{P}\left( \min_{1 \leq j \leq m} a_{ij} > y \right) \tag{13}$$

$$= 1 - \mathbb{P}\left( \bigcap_j a_{ij} > y \right) = 1 - (1 - F_D(y)^m) = F_D(y)^m. \tag{14}$$

Next, we consider $Z := \max_{1 \leq i \leq n} \delta_i$. To evaluate the density of $Z$, we first need to derive its distribution $F_Z$:

$$F_Z(z) = \mathbb{P}(Z \leq z) = \mathbb{P}\left( \max_{m < i \leq n} \delta_i \leq z \right) = \mathbb{P}\left( \bigcap_{m < i \leq n} \delta_i \leq z \right) \tag{15}$$

Next, we approximate $Z$ by $Z'$ by imposing *i.i.d sampling* of the minimal distances $\delta_i$ from $F_\Delta$. This is an approximation because in practice, the rows $m + 1$ to $n$ are not stochastically independent because of the triangular inequality that holds for metrics. However, assuming i.i.d., we arrive at

$$F_{Z'}(z) = F_\Delta(z)^{n-m} \tag{16}$$

. Since $Z'$ has positive support its expectation can then be evaluated as:

$$\mathbb{E}_{Z' \sim F_{Z'}}[Z'] = \int_0^{+\infty} (1 - F_{Z'}(z)) \, \mathrm{d}z = \int_0^{+\infty} \left(1 - F_\Delta(z)^{n-m}\right) \mathrm{d}z \tag{17}$$

$$= \int_0^{+\infty} \left(1 - F_D(z)^{m(n-m)}\right) \mathrm{d}z \geq \int_0^{+\infty} \left(1 - F_D(z)^{(n-1)}\right) \mathrm{d}z \tag{18}$$

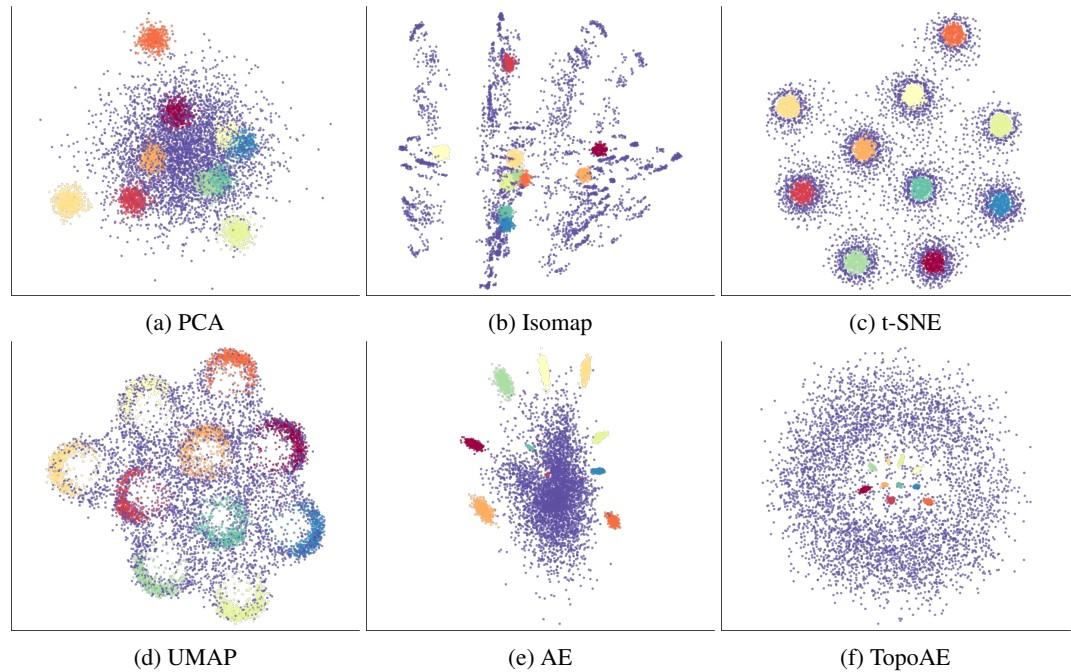

(a) PCA  (b) Isomap  (c) t-SNE

(d) UMAP  (e) AE  (f) TopoAE

Figure A.2: A depiction of *all* latent spaces obtained for the SPHERES data set. TopoAE used batch-size 28. This is an enlarged version of the figure shown in Section 5.2.

The independence assumption leading to $Z'$ results in *overestimating* the variance of the drawn minima $\delta_i$. Thus, the expected maximum of those minima, $\mathbb{E}[Z']$, is overestimating the actual expectation of the maximum $\mathbb{E}[Z]$, which is why Eq. 17 to Eq. 18 constitute an *upper bound* of $\mathbb{E}[Z]$, and equivalently, an upper bound of $\mathbb{E}\big[d_H(X, X^{(m)})\big]$. When increasing $m$, $\mathbb{E}[d_H(X, X^m)]$ decreases monotonically since for a particular $m$, we draw $n - m$ samples from the minimal distance distribution $F_\Delta$, and their maximum determines the Hausdorff distance. In contrast, our preliminary upper bound on the left-hand side of Eq. 18 forms a downwards-facing parabola due to the quadratic form in the exponent. This indicates that a tighter bound is achieved for $m \neq n$ by using the minimal subsample size of $m = 1$.

$\square$

## A.4  SYNTHETIC DATASET

SPHERES consists of eleven high-dimensional 100-spheres living in $101-$dimensional space. Ten spheres of radius $r = 5$ are each shifted in a random direction (by adding the same Gaussian noise vector per sphere). To this end, we draw ten $d$-dimensional Gaussian vectors following $\mathcal{N}(\mathbf{0}, \mathbf{I}(10/\sqrt{d}))$ for $d = 101$. Crucially, to add interesting topological information to the data set, the ten spheres are enclosed by an additional larger sphere of radius $5r$. The spheres were generated using the library `scikit-tda`.

## A.5  ARCHITECTURES AND HYPERPARAMETER TUNING

**Architectures for synthetic data set**  For the synthetically generated dataset we use a simple Multilayer perceptron architecture consisting of two hidden layer with 32 neurons each both encoder and decoder and a bottleneck of two neurons such that the sequence of hidden-layer neurons is $32 - 32 - 2 - 32 - 32$. ReLU non-linearities and batch normalization were applied between the layers excluding the output layer and the bottleneck layer. The networks were fit using mean squared error loss.

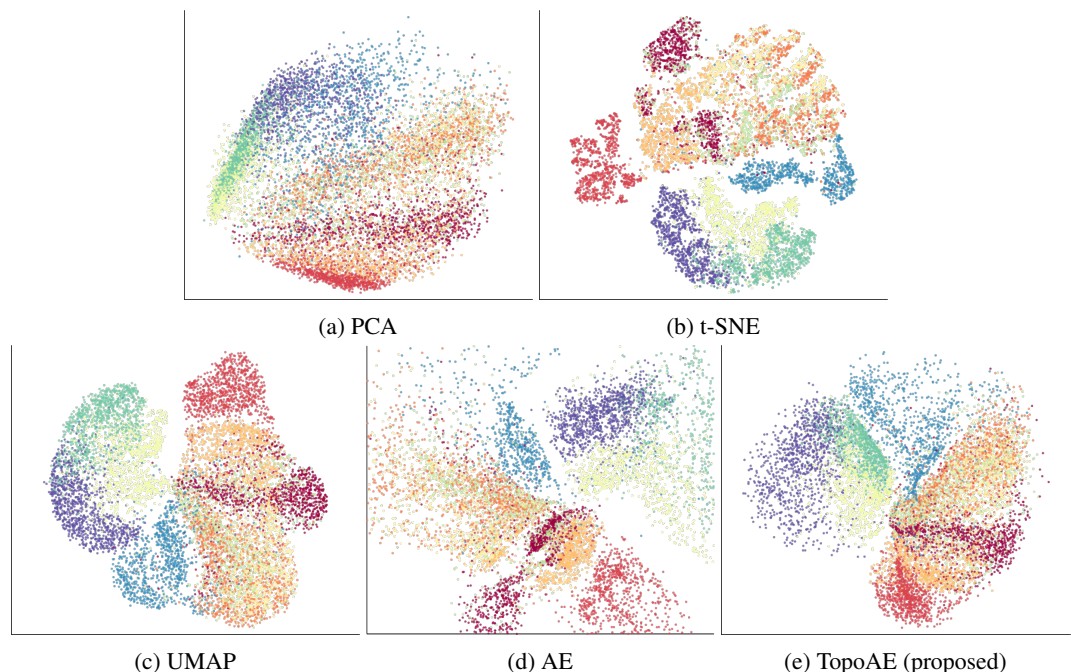

Figure A.3: Latent representations of the FASHION-MNIST data set. TopoAE used batch-size 95. This is a larger extension of the figure shown in Section 5.2.

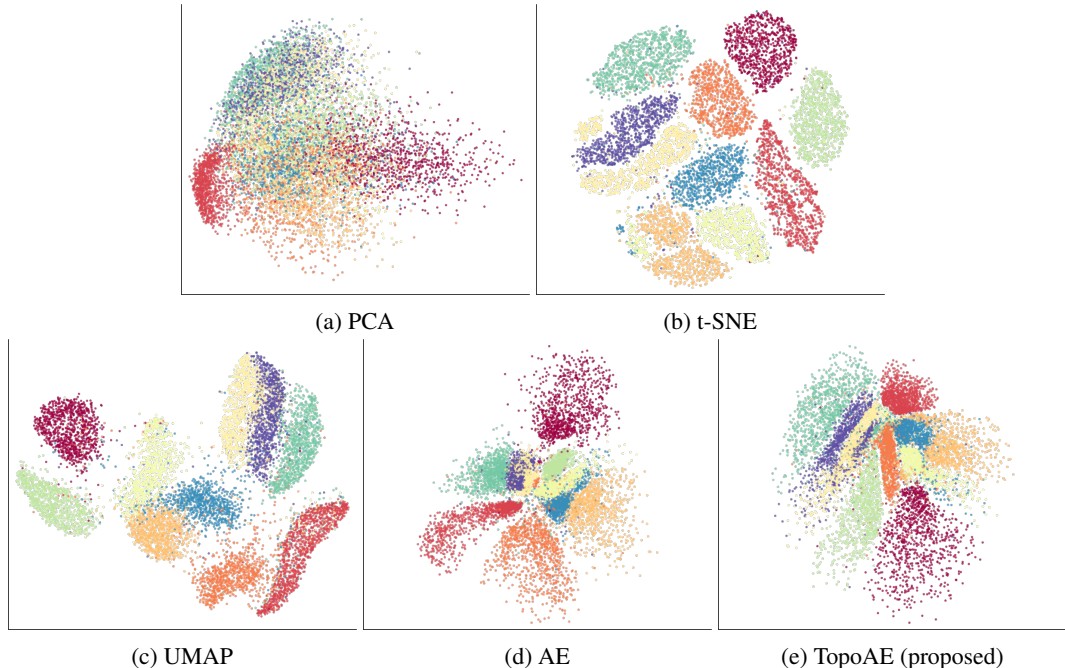

Figure A.4: Latent representations of the MNIST data set. TopoAE used batch-size 126. This is a larger extension of the figure shown in Section 5.2.

**Architectures for real world data sets** For the MNIST, FASHION-MNIST, and CIFAR-10 datasets, we use an architecture inspired by DeepAE (Hinton & Salakhutdinov, 2006). This architecture is composed of 3 layers of hidden neurons of decreasing size $(1000 - 500 - 250)$ for the encoder part, a bottleneck of two neurons, and a sequence of three layers of hidden neurons in decreasing size

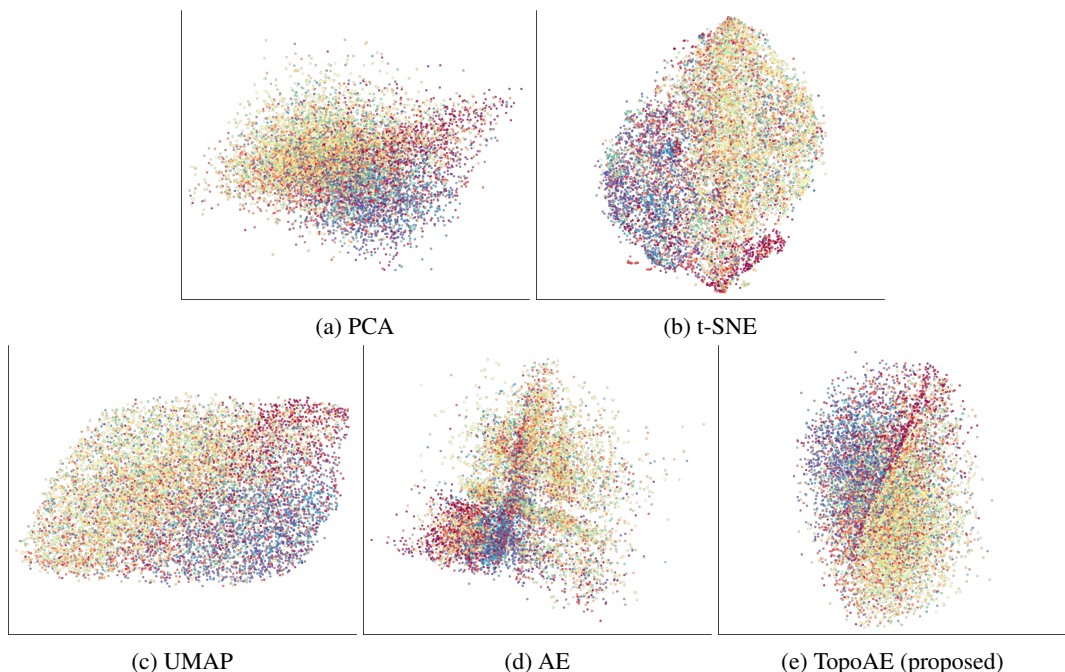

(a) PCA           (b) t-SNE

(c) UMAP       (d) AE       (e) TopoAE (proposed)

Figure A.5: Latent representations of the CIFAR-10 data set. TopoAE used batch-size 82. This is a larger extension of the figure shown in Section 5.2.

$(250 - 500 - 1000)$ for the decoder. In contrast to the originally proposed architecture, we applied ReLU non-linearities and batch normalization between the layers as we observed faster and more stable training. For the non-linearities of the final layer, we applied the $tanh$ non-linearity, such that the image of the activation matches the range of input images scaled between $-1$ and $1$. Also here, we applied mean squared error loss.

All neural network architectures were fit using Adam and weight-decay of $10^{-5}$.

**Hyperparameter tuning** For hyperparameter tuning we apply random sampling of hyperparameters using the `scikit-optimize` library (scikit-optimize contributers, 2018) with 20 calls per method on all datasets. We select the best model parameters in terms of $\mathrm{KL}_{0.1}$ on the validation split and evaluate and report it on the test split. To estimate performance means and standard deviations, we repeated the evaluation on an independent test split 5 times by using the best parameters (as identified in the hyperparameter search on the validation split) and refitting the models by resampling the train-validation split.

**Neural networks** For the neural networks we sample the learning rate log-uniformly in the range $10^{-4} - 10^{-2}$, the batch-size uniformly between 16 and 128, and for the TopoAE method we sample the regularization strength log-uniformly in the range $10^{-1} - 3$. Each model was allowed to train for at most 100 epochs, whereas we applied early stopping with patience $= 10$ based on the validation loss.

**Competitor methods** For t-SNE, we sample the perplexity uniformly in the range $5 - 50$ and the learning rate log-uniformly in the range $10 - 1000$. For Isomap and UMAP, the number of neighbors included in the computation was varied between $15 - 500$. For UMAP, we additionally vary the min_dist parameter uniformly between 0 and 1.

### A.6 MEASURING THE QUALITY OF LATENT REPRESENTATIONS

Next to the reconstruction error (if available; please see the paper for a discussion on this), we use a variety of NLDR metrics to assess the quality of our method. Our primary interest concerns

the quality of the latent space because, among others, it can be used to visualise the data set. We initially considered classical quality metrics from non-linear dimensionality reduction (NLDR) algorithms (see Bibal & Frénay (2019); Gracia et al. (2014); van der Maaten et al. (2009) for more in-depth descriptions), namely

(1) the *root mean square error* ($\ell$-RMSE) between the distance matrix of the original space and the latent space (as mentioned in the main text, this is not related to the reconstruction error),
(2) the *mean relative rank error* ($\ell$-MRRE), which measures the changes in *ranks* of distances in the original space and the latent space (Lee & Verleysen, 2009),
(3) the *trustworthiness* ($\ell$-Trust) measure (Venna & Kaski, 2006), which checks to what extent the $k$ nearest neighbours of a point are preserved when going from the original space to the latent space, and
(4) the *continuity* ($\ell$-Cont) measure (Venna & Kaski, 2006), which is defined analogously to $\ell$-Trust, but checks to what extent neighbours are preserved when going from the *latent* space to the original space.

All of these measures are defined based on comparisons of the original space and the latent space; the reconstructed space is *not* used here. As an additional measure, we calculate the *Kullback–Leibler divergence* between density distributions of the input space and the latent space. Specifically, for a point cloud $X$ with an associated distance $\mathrm{dist}$, we first use the *distance to a measure* density estimator (Chazal et al., 2011; 2014b), defined as $\mathrm{f}_\sigma^{\mathcal{X}}(x) := \sum_{y \in \mathcal{X}} \exp\left(-\sigma^{-1} \mathrm{dist}(x,y)^2\right)$, where $\sigma \in \mathbb{R}_{>0}$ represents a length scale parameter. For $\mathrm{dist}$, we use the Euclidean distance and normalise it between $0$ and $1$. Given $\sigma$, we evaluate $\mathrm{KL}_\sigma := \mathrm{KL}\left(\mathrm{f}_\sigma^X \parallel \mathrm{f}_\sigma^Z\right)$, which measures the similarity between the two density distributions. Ideally, we want the two distributions to be similar because this implies that density estimates in a low-dimensional representation are similar to the ones in the original space.

## A.7 Assessing the batch size

As we used fixed architectures for the hyperparameter search, the batch-size remains the main determinant for the runtime of TopoAE. In Figure A.6, we display trends (linear fits) on how loss measures vary with batch size. Addtionally, we draw runtime estimates. As we applied early stopping, for better comparability, we approximated the epoch-wise runtime by dividing the execution time of a run by its number of completed epochs. Interestingly, these plots suggest that the runtime grows with decreasing batch-size (even though the topological computation is more costly for larger batch-sizes!). In these experiments, sticking to $0-$dimensional topological features we conclude that the benefit of using mini-batches for neural network training still dominate the topological computations. The few steep peaks most likely represent outliers (the corresponding runs stopped after few epochs, which is why the effective runtime could be overestimated).

For the loss measures, we see that reconstruction loss tends to *decrease* with increasing batch-size, while our topological loss tends to *increase* with increasing batch-size (despite normalization). The second observation might be due to larger batch-size enabling more complex data point arrangements and corresponding topologies.

## A.8 Extending to variational autoencoders

In Figure A.7 we sketch a preliminary experiment, where we apply our topological constraint to variational autoencoders for the SPHERES dataset. Also here, we observe that our constraint helps identifying the nesting structure of the enclosing sphere.

## A.9 Topological distance calculations

To assess the topological fidelity of the resulting latent spaces, we calculate several topological distances between the test data set (full dimensionality) and the latent spaces obtained from each method (two dimensions). More precisely, we calculate (i) the $1^{\mathrm{st}}$ Wasserstein distance ($W_1$), (ii) the $2^{\mathrm{nd}}$ Wasserstein distance ($W_2$), and (iii) the bottleneck distance ($W_\infty$) between the persistence diagrams obtained from the test data set of the SPHERES data and their resulting 2D latent representations. Even though our loss function is *not* optimising this distance, we observe in Table A.1 that

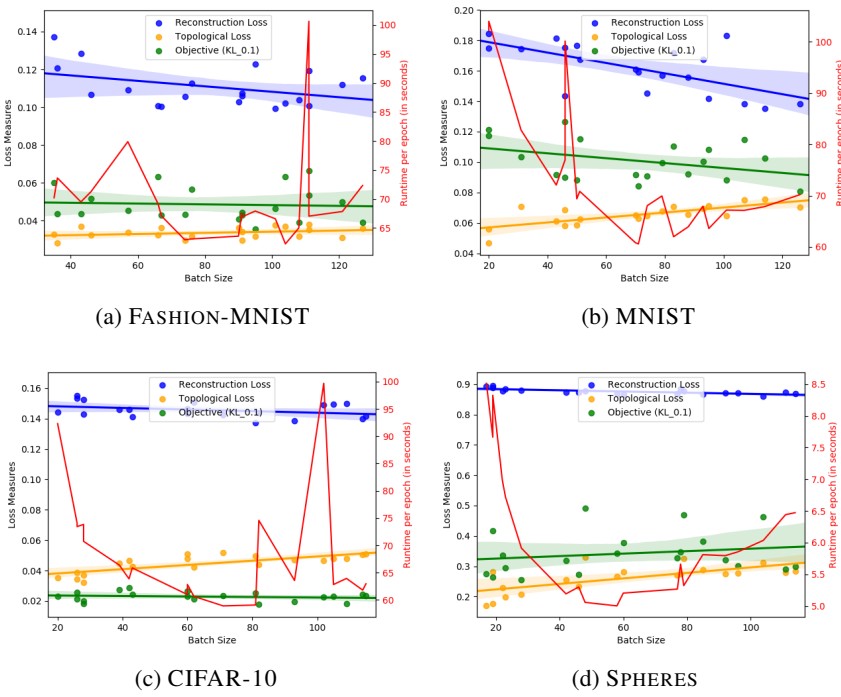

(a) FASHION-MNIST

(b) MNIST

(c) CIFAR-10

(d) SPHERES

Figure A.6: From our hyperparameter searches we scatter batch-sizes against three measures of interest: Topological Loss, Reconstruction Loss, and $KL_{0.1}$, our objective for hyperparameter search. Additionally, we draw epoch-wise runtime estimates.

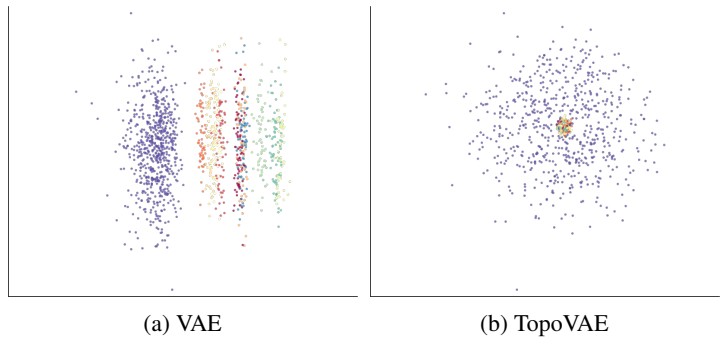

(a) VAE

(b) TopoVAE

Figure A.7: A depiction of latent spaces obtained for the SPHERES data set with variational autoencoders (VAEs). Here, VAE represents a standard MLP-based VAE, whereas TopoVAE represents the same architecture plus our topological constraint.

Table A.1: Various topological distances calculated between the test data set and the corresponding latent space. For computational efficiency reasons, we used subsamples of size $m = 500$ and repeated the process 10 times. Individual cells thus contain a mean and a standard deviation.

| Method | $W_1$ | $W_2$ | $W_\infty$ |
|--------|-------|-------|------------|
| Isomap | 4.32±0.037 | 0.477±0.0045 | 0.165±0.00096 |
| PCA | 4.42±0.053 | 0.476±0.0046 | 0.158±0.00108 |
| t-SNE | 4.38±0.038 | 0.478±0.0045 | 0.164±0.00094 |
| UMAP | 4.47±0.042 | 0.478±0.0045 | 0.160±0.00092 |
| AE | 3.99±0.037 | 0.469±0.0053 | 0.154±0.00128 |
| TopoAE | 3.73±0.076 | 0.459±0.0055 | 0.152±0.00268 |

the topological distance of our method ("TopoAE") is always the lowest among all the methods. In particular, it is *always* smaller than the topological distance of the latent space of the autoencoder architecture; this is true for all distance measures, even though $W_\infty$, for example, is known to be susceptible to outliers. Said experiment serves as a simple "sanity check" as it demonstrates that the changes induced by our method are beneficial in that they reduce the topological distance of the latent space to the original data set. For a proper comparison of topological features between the two sets of spaces, a more involved approach would be required, though.

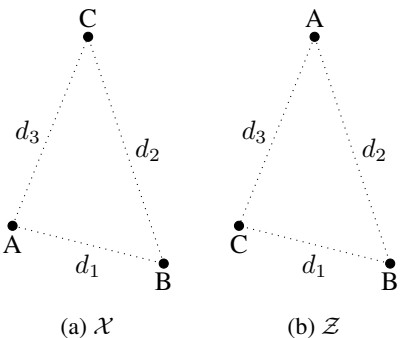

(a) $\mathcal{X}$        (b) $\mathcal{Z}$

Figure A.8: An undesirable configuration of the latent space of three non-collinear points. Pairwise distances are shown as dotted lines. This configuration results in equal persistence diagrams for $\mathcal{X}$ and $\mathcal{Z}$. We prevent this by not *explicitly* minimising the distances between persistence diagrams but by instead including the persistence pairing information.

### A.10 ALTERNATIVE LOSS FORMULATIONS

Our choice of loss function was motivated by the observation that *only* aligning the persistence diagrams between mini-batches of $\mathcal{X}$ and $\mathcal{Z}$ can lead to degenerate or 'meaningless' latent spaces. As a simple example (see Figure A.8 for a visualisation), imagine three non-collinear points in the input space and the triangle they are forming. Now assume that the latent space consists of the same triangle (in terms of its side lengths) but with permuted labels. A loss term of the form

$$\mathcal{L}' := \left\| \mathbf{A}^X \left[ \pi^X \right] - \mathbf{A}^Z \left[ \pi^Z \right] \right\|^2 \tag{19}$$

only measures the distance between persistence diagrams (which would be zero in this situation) and would not be able to penalise such a configuration.

Table A.2: Extended version of the table from the main paper, showing more length scales and variance estimates.

| Data set | Method | $KL_{0.001}$ | $KL_{0.01}$ | $KL_{0.1}$ | $KL_1$ | $KL_{10}$ | $\ell$-Cont | $\ell$-MRRE | $\ell$-Trust | $\ell$-RMSE | Data MSE |
|---|---|---|---|---|---|---|---|---|---|---|---|
| SPHERES | Isomap | $0.53095 \pm 0.01929$ | $0.18096 \pm 0.02547$ | $0.42048 \pm 0.00559$ | $0.00881 \pm 0.00020$ | $0.00009 \pm 0.00000$ | $0.79027 \pm 0.00244$ | $0.24573 \pm 0.00158$ | $0.67643 \pm 0.00323$ | $10.37188 \pm 0.22856$ | – |
| | PCA | $0.22445 \pm 0.00691$ | $0.33231 \pm 0.00552$ | $0.65121 \pm 0.00256$ | $0.01530 \pm 0.00010$ | $0.00016 \pm 0.00000$ | $0.74740 \pm 0.00140$ | $0.29402 \pm 0.00108$ | $0.62557 \pm 0.00066$ | $11.76482 \pm 0.01460$ | $0.96103 \pm 0.00029$ |
| | TSNE | $0.22794 \pm 0.00722$ | $0.15228 \pm 0.00805$ | $0.52722 \pm 0.03261$ | $0.01271 \pm 0.00058$ | $0.00013 \pm 0.00001$ | $0.77300 \pm 0.00513$ | $0.21740 \pm 0.00472$ | $0.67862 \pm 0.00474$ | $8.05018 \pm 0.11057$ | – |
| | UMAP | $0.24752 \pm 0.01917$ | $0.15687 \pm 0.00599$ | $0.61326 \pm 0.00752$ | $0.01658 \pm 0.00028$ | $0.00018 \pm 0.00000$ | $0.75153 \pm 0.00360$ | $0.24968 \pm 0.00094$ | $0.63483 \pm 0.00185$ | $9.27009 \pm 0.03417$ | – |
| | AE | $0.28432 \pm 0.02165$ | $0.56571 \pm 0.02864$ | $0.74588 \pm 0.04323$ | $0.01664 \pm 0.00115$ | $0.00017 \pm 0.00001$ | $0.60663 \pm 0.01685$ | $0.34918 \pm 0.00903$ | $0.58843 \pm 0.00475$ | $13.33061 \pm 0.05198$ | $0.81545 \pm 0.00106$ |
| | TopoAE | $0.62765 \pm 0.05415$ | $0.08504 \pm 0.01270$ | $0.32572 \pm 0.02050$ | $0.00694 \pm 0.00055$ | $0.00007 \pm 0.00001$ | $0.82200 \pm 0.01813$ | $0.27239 \pm 0.01108$ | $0.65775 \pm 0.01428$ | $13.45753 \pm 0.04177$ | $0.86812 \pm 0.00074$ |
| FASHION-MNIST | PCA | $0.22559 \pm 0.00011$ | $0.35594 \pm 0.00004$ | $0.05205 \pm 0.00004$ | $0.00069 \pm 0.00000$ | $0.00001 \pm 0.00000$ | $0.96777 \pm 0.00001$ | $0.05744 \pm 0.00001$ | $0.91681 \pm 0.00003$ | $9.05121 \pm 0.00041$ | $0.18439 \pm 0.00000$ |
| | TSNE | $0.03516 \pm 0.00226$ | $0.40477 \pm 0.01251$ | $0.07095 \pm 0.00962$ | $0.00198 \pm 0.00026$ | $0.00002 \pm 0.00000$ | $0.96731 \pm 0.00268$ | $0.01962 \pm 0.00073$ | $0.97405 \pm 0.00070$ | $41.25460 \pm 0.53671$ | – |
| | UMAP | $0.05069 \pm 0.00238$ | $0.42362 \pm 0.00609$ | $0.06491 \pm 0.00161$ | $0.00163 \pm 0.00005$ | $0.00002 \pm 0.00000$ | $0.98126 \pm 0.00016$ | $0.02867 \pm 0.00034$ | $0.95874 \pm 0.00060$ | $13.68933 \pm 0.02896$ | – |
| | AE | $0.17177 \pm 0.13603$ | $0.47798 \pm 0.09567$ | $0.06791 \pm 0.00700$ | $0.00125 \pm 0.00017$ | $0.00001 \pm 0.00000$ | $0.96849 \pm 0.00372$ | $0.02562 \pm 0.00217$ | $0.97418 \pm 0.00119$ | $20.70674 \pm 3.56861$ | $0.10197 \pm 0.00222$ |
| | TopoAE | $0.11039 \pm 0.02948$ | $0.39204 \pm 0.03264$ | $0.05353 \pm 0.00959$ | $0.00100 \pm 0.00015$ | $0.00001 \pm 0.00000$ | $0.97998 \pm 0.00194$ | $0.03156 \pm 0.00253$ | $0.95612 \pm 0.00391$ | $20.49122 \pm 0.93206$ | $0.12071 \pm 0.00238$ |
| MNIST | PCA | $0.16754 \pm 0.00051$ | $0.38876 \pm 0.00146$ | $0.16301 \pm 0.00059$ | $0.00160 \pm 0.00001$ | $0.00002 \pm 0.00000$ | $0.90084 \pm 0.00016$ | $0.16582 \pm 0.00022$ | $0.74546 \pm 0.00048$ | $13.17437 \pm 0.00216$ | $0.22269 \pm 0.00002$ |
| | TSNE | $0.03767 \pm 0.00140$ | $0.27695 \pm 0.05266$ | $0.13266 \pm 0.02362$ | $0.00214 \pm 0.00041$ | $0.00002 \pm 0.00000$ | $0.92101 \pm 0.00288$ | $0.03953 \pm 0.00129$ | $0.94624 \pm 0.00147$ | $22.89261 \pm 0.24373$ | – |
| | UMAP | $0.07214 \pm 0.00091$ | $0.32063 \pm 0.00320$ | $0.14568 \pm 0.00207$ | $0.00234 \pm 0.00004$ | $0.00003 \pm 0.00000$ | $0.93992 \pm 0.00066$ | $0.05109 \pm 0.00022$ | $0.93770 \pm 0.00039$ | $14.61535 \pm 0.04332$ | – |
| | AE | $0.44690 \pm 0.08540$ | $0.61993 \pm 0.11742$ | $0.15542 \pm 0.02203$ | $0.00156 \pm 0.00023$ | $0.00002 \pm 0.00000$ | $0.91293 \pm 0.00564$ | $0.05828 \pm 0.00353$ | $0.93699 \pm 0.00262$ | $18.18105 \pm 0.21459$ | $0.13732 \pm 0.00160$ |
| | TopoAE | $0.32427 \pm 0.03312$ | $0.34069 \pm 0.03056$ | $0.11012 \pm 0.01069$ | $0.00114 \pm 0.00010$ | $0.00001 \pm 0.00000$ | $0.93210 \pm 0.00132$ | $0.05553 \pm 0.00044$ | $0.92844 \pm 0.00142$ | $19.57784 \pm 0.01812$ | $0.13884 \pm 0.00066$ |
| CIFAR | PCA | $0.27320 \pm 0.00014$ | $0.59073 \pm 0.00004$ | $0.01961 \pm 0.00001$ | $0.00023 \pm 0.00000$ | $0.00000 \pm 0.00000$ | $0.93130 \pm 0.00000$ | $0.11921 \pm 0.00005$ | $0.82117 \pm 0.00002$ | $17.71567 \pm 0.00084$ | $0.14816 \pm 0.00000$ |
| | TSNE | $0.04451 \pm 0.00222$ | $0.62733 \pm 0.01427$ | $0.03014 \pm 0.00333$ | $0.00073 \pm 0.00007$ | $0.00001 \pm 0.00000$ | $0.90300 \pm 0.00611$ | $0.10265 \pm 0.00242$ | $0.86325 \pm 0.00151$ | $25.61099 \pm 0.11551$ | – |
| | UMAP | $0.06934 \pm 0.00202$ | $0.61673 \pm 0.00052$ | $0.02562 \pm 0.00019$ | $0.00050 \pm 0.00001$ | $0.00001 \pm 0.00000$ | $0.92045 \pm 0.00013$ | $0.12680 \pm 0.00028$ | $0.81668 \pm 0.00019$ | $33.57785 \pm 0.00796$ | – |
| | AE | $0.37737 \pm 0.06507$ | $0.66834 \pm 0.02992$ | $0.03458 \pm 0.00448$ | $0.00062 \pm 0.00021$ | $0.00001 \pm 0.00000$ | $0.85072 \pm 0.00429$ | $0.13204 \pm 0.00316$ | $0.86359 \pm 0.00442$ | $36.26827 \pm 0.56159$ | $0.14030 \pm 0.00190$ |
| | TopoAE | $0.20877 \pm 0.00951$ | $0.55642 \pm 0.00412$ | $0.01879 \pm 0.00051$ | $0.00031 \pm 0.00002$ | $0.00000 \pm 0.00000$ | $0.92691 \pm 0.00100$ | $0.10809 \pm 0.00210$ | $0.84514 \pm 0.00359$ | $37.85914 \pm 0.03303$ | $0.13975 \pm 0.00171$ |

