# OpenReview forum: "Topological Autoencoders"
_ICLR.cc/2020/Conference — Reject_

### Official Review · AnonReviewer1 · 2019-10-22
**Official Blind Review #1**

**Rating:** 6

**Review:**

Summary
-------------

The paper proposes an approach, based on persistent homology (PH), to preserve certain topological structures of the input in the latent representations learned by an autoencoder. This is realized via an additional (i.e., in addition to reconstruction) loss term (optimized over mini-batches) which requires differentiating through the PH computation. While this has been done before (e.g., Chen et al., Hofer et al.), the authors have certainly put an interesting spin on this. The theoretical part of the work deals with the issue of using mini-batches for PH computation and whether this computation is close to the computation on the full point cloud. Experiments and comparisons on multiple datasets are presented to demonstrate that the approach, e.g., preserves nesting relationships in the input. The paper is nicely written and the content is very well presented. There are questions here and there (see below), but I do think they can be answered.

Major comments/remarks:
-------------------------------------

My first question relates to the issue that the loss only incorporates 0-dim. information. The authors do remark that higher-dim. features can be included, but the results were similar. However, after thinking about this issue quite some time, I am curious if it is possible to obtain "zero" of the topological loss (so this term is perfectly optimized), but the encoder introduces, e.g., cavities in the data which were not present in the input (e.g., 1-dim. holes).

Also, can you show formally (maybe this is trivial and I am not seeing it) that L_t = 0 would lead to 0 distance between the corresponding diagrams w.r.t. some common metric? A more formal treatment of the implications of the loss in Eq. (2) would certainly help.

Another question that immediately comes to mind is whether the computation of VR PH in the input space (e.g., CIFAR 10) makes sense, as the authors rely on ||.||_2 if I understood this correctly. I would argue that the topology of the input is basically unknown, especially for images and computing Euclidean distances among images, or vectorized images, does not make sense. For the nice results on the SPHERES data set it does, as the spheres are defined exactly using ||.||_2. If the VR PH in 0-dim. of the input is enforced upon the representations in the AE bottleneck, but the input topology is not captured well, then you might be enforcing something that you possibly do not want.

Apart from that, it is known that the Euclidean distance degenerates quickly in high dimensional spaces, e.g.,

Aggrawal et al.
On the Surprising Behavior of Distance Metrics in High Dimensional Space

Maybe this is also contributing to the fuzzy visualization of CIFAR-10 in Fig. 3 (apart from the low-dim. of the bottleneck)?

Also, maybe the authors could work out (in greater detail) the differences between their results from Thm.1/2 and the results of Chazal et al., in "Subsampling Methods for Persistent Homology". In my point of view, the results in the paper only hold if you would consider just a single batch, right? I mean, if the loss is computed from the batch, and a gradient update is performend, Z^{m} will changes (as the encoder changes as a result of the update), while the input does not.

Finally, how were the KL divergence measures in Table 1 computed, as you need a density estimate of the input as well, not just for the representation space, right? Is this not a very crucial issue in the input space? If so, how reliable are the numbers presented for KL_{0.01},etc., given that the differences are sometimes extremely small.

Minor comments
-----------------------

Sec. 6: We presented a topological autoencoders -> We presented a topological autoencoder

Overall, I think this is a nicely done paper, but with quite some question marks at many places. I do think this is always the case for something new, though, and
actually a good thing.

**Experience Assessment:**

I have published in this field for several years.

**Review Assessment: Checking Correctness Of Derivations And Theory:**

I carefully checked the derivations and theory.

**Review Assessment: Checking Correctness Of Experiments:**

I assessed the sensibility of the experiments.

**Review Assessment: Thoroughness In Paper Reading:**

I read the paper thoroughly.

---

> ### Author Response · Authors · 2019-11-08
> **Our comments to your review**
>
> > [...] the loss only incorporates 0-dim. information [...]
>
> We restricted the scope of the calculations in this paper. Our loss can generally be computed for higher-dimensional topological features as well. This requires pairing a given edge of the Vietoris--Rips complex with the higher-order simplex it is creating or destroying during filtration. This works because the Vietoris--Rips complex is a clique complex, i.e. its structure is completely determined by its edges. For this paper, we implemented dimensions 0 and 1; for the data sets that we analysed, we saw no benefits in 1-dimensional features; only the runtime increased. This is not a universal claim, though, and we believe that there are other data sets that exist that require higher-dimensional topological features.
>
> >  [...] but the encoder introduces, e.g., cavities in the data
>
> An autoencoder might possibly introduce higher-order topological structures to the latent space if not being “punished” for doing so; however, AEs are also not incentivised to do so,
> which might be why we did not observe the value of adding 1-dimensional features. Interestingly, there are symmetry properties of PH that lead to some redundant information in  higher-dimensional features; this restricts the potential of AEs to introduce additional information without noticing it in lower-dimensional features. Please see https://dx.doi.org/10.1007/s10208-008-9027-z for more details.
>
> > L_t = 0 [...]
>
> Thanks for pointing this out! If the persistence diagrams and pairings are the same, $L_t =0$, provided the distances are scaled properly. The converse does not necessarily hold. We now mention this in the paper. We do not merely try to arrive at identical entries in the persistence diagrams (irrespective of the underlying pairing), which could be achieved by minimising the distance $\|A^X[\pi^X] - A^Z[\pi^Z]\|$, for example. For such a loss term, which disregards the minibatch sample indices, zero loss would imply equal persistence diagrams; however, the loss would not punish “meaningless” latent spaces that do not represent samples correctly. For example, imagine a minibatch of 3 non-collinear samples forming a triangle in data space (see A.10). If the latent space shows the exact same triangle shape but with permuted labels (sample indices), the alternative loss term (oblivious to the pairings) would not punish this.
>
> > Use of Euclidean distance
>
> For reasons of clarity and to harmonise our setup, we used the Euclidean distance in the paper. The choice of distance influences to what extent PH is able to capture topological features—it is therefore crucial, but typically, suitable results can be achieved with Euclidean distances. However, our method supports different distances, which we now discuss more prominently in the paper. For higher dimensions, it would be interesting to employ fractional metrics instead. Iit would also be possible to compute the distance using activations of intermediate layers of pretrained convolutional neural networks in order to derive a “more meaningful” distance metric. We consider these promising directions for future work. In this paper, we focused on adjusting the topologies between input and latent space without introducing another layer of complexity.
>
> > Fuzzy visualization of CIFAR-10
>
> In our experiments, CIFAR-10 proved to be challenging to directly compress to 2D as compared to weaker bottlenecks (e.g. 100 dimensions) followed by t-SNE. Since our aim was dimensionality reduction and visualization, we uniformly chose two latent dimensions for all data sets.
>
> > Thm. 1/2
>
> Both theorems prove that subsampling of a point cloud is valid, insofar as it still captures similarity between persistence diagrams well. Our theoretical setup is simpler than that of Chazal et al., who discuss subsampling from a measure-theoretical point of view. Thm. 1 is a proof of the stability of the PH calculation at the minibatch level, while Thm. 2 bounds  the distances of persistence diagrams as the subsample size $m$ increases. Both theorems discuss the robustness of point cloud subsampling irrespective of belonging to the input space or the latent space. That is, even though the latent space changes during training, at each step during training, the PH of the latent minibatch $Z^{m}$ is a robust approximation of the PH of the full data set in latent space (at this moment in training).
>
> > KL divergence
>
> We discuss the details of KL divergence calculations in Section A.6. The computation requires a base distance (where we use again the Euclidean distance; please also see the point above), which might be susceptible to noise. We follow the setup of Chazal et al. (2011, 2014b), but it will be interesting to consider the impact of other distances in future work.
>
> > Typos in Section 6
>
> We also rewrote Section 6 and corrected typos. Thanks!
>
> Please let us know if there are any other questions that we can answer for you concerning this paper.

---

### Official Review · AnonReviewer2 · 2019-10-22
**Official Blind Review #2**

**Rating:** 8

**Review:**

This paper shows how to train an autoencoder that preserves topologically
relevant distance across varying length scales.  It presents a new
differentiable loss term for topological distance between input and latent
space.  The topological loss has a number of nice features: compatibility with
minibatch training, and extension to higher input and latent space
dimensionality fairly easily.

Since this trainable topological loss can be applied in more general scenarios
than simply auto-encoders or visualization, I think this well-written article
is of general interest and worth publishing.

The main ideas and related work are presented clearly in sections 1--4, and the
experiments compare TopoAE with a variety of low-dimensional visualization
methods.  Embedding quality is evaluated with a wide variety of metrics on real
and synthetic datasets highlighting the preservation of global and local
topology.

My only real issue, easily fixable, was that after the minibatch stability
theory (Sec. 3.3), I really wanted to know what minibatch sizes were used in
the experimental figures, and spent twenty minutes before admitting defeat.
Please include the actual TopoAE minibatch size when presenting the
visualization figures in the article and appendices.

The appendices contain a wealth of useful reading and experiments, and the
provided source code was clearly organized and useful to browse.  I did not go
line-by-line through the appendix proofs.

**Experience Assessment:**

I have published one or two papers in this area.

**Review Assessment: Checking Correctness Of Derivations And Theory:**

I assessed the sensibility of the derivations and theory.

**Review Assessment: Checking Correctness Of Experiments:**

I assessed the sensibility of the experiments.

**Review Assessment: Thoroughness In Paper Reading:**

I read the paper thoroughly.

---

> ### Author Response · Authors · 2019-11-08
> **Our comments to your review**
>
> Thank you very much for your encouraging review! Following your suggestion, we now added the TopoAE mini-batch sizes to all visualisations in the text. Please let us know if there are any other questions that we can answer for you concerning this paper.

---

### Official Review · AnonReviewer3 · 2019-10-27
**Official Blind Review #3**

**Rating:** 3

**Review:**

The basic premise of this work is that topological operators are increasingly adopted in machine learning models, but the literature lacks of general differentiable topological operators.

The main idea proposed in this work is to use the topological signature directly as a loss for autoencoders. The general aim is to preserve the topological properties of data while performing dimensionality reduction.

The proposed method computes the persistent homology of the 0-dimensional Vietoris–Rips by means of persistence diagrams and persistence pairing. The additional step is to computes the regularization loss, by posing the constraint that the persistent homology be similar between data space and latent space. The empirical analysis investigates the stability property with respect to the mini-batch sampling.

The extended experiments don't provide an empirical evidence of the added value of a topological AE. The results are quite controversial if we focus our attention on MSE measures. AE and TopoAE behave in a similar way and it is not clear whether such a difference is statistically significant.

The work is missing a mandatory comparison with Hofer et al. (2019b) that proposes a loss very similar to the one presented in Section 3, even though the authors remark that their formulation is more general. For example an interesting comparison could have been with the real-case application of classification for the images datasets (e.g. CIFAR, MNIST, ImageNet) where Hofer et al. (2019b) proves their claims.

The computational issues of the d-dimensional Vietoris-Rips with d > 0, limits the preservation of only simple structures in the point-cloud.


**Experience Assessment:**

I have read many papers in this area.

**Review Assessment: Checking Correctness Of Derivations And Theory:**

I assessed the sensibility of the derivations and theory.

**Review Assessment: Checking Correctness Of Experiments:**

I assessed the sensibility of the experiments.

**Review Assessment: Thoroughness In Paper Reading:**

I read the paper at least twice and used my best judgement in assessing the paper.

---

> ### Author Response · Authors · 2019-11-08
> **Our comments to your review**
>
> > MSE to compare AE and Topo-AE
>
> We understand the concerns of the reviewer here. To clarify, our method regularises the topology of the latent space as an *additional* objective. The goal is to obtain a latent space whose topology is similar to that of the input space (as approximated on the level of mini-batches). Therefore, we consider the small increase in MSE to be the ‘price to pay’ for obtaining a better (in terms of its topology) latent representation. To illustrate this point, imposing structural restrictions at the minimal cost of increased reconstruction error is comparable to the relationship between SOM and the standard k-means approach.
>
> > The work is missing a mandatory comparison with Hofer et al. (2019b)
>
> Thank you for pointing this out. While our work is related to Hofer et al. (2019b) the objective of this work is different. Hofer et al. propose regularizing the connectivity of the latent space by itself, whereas our goal is to align input and latent space in terms of connectivity. In particular Hofer et al. show that it could be beneficial for some downstream task (such as classification) to enforce certain topological properties in a representation and allow to tune these properties in the latent space. By contrast, we propose a method for dimensionality reduction which should *preserve* the topology of a data space in the latent space, to allow a better understanding and visualization of the high-dimensional data space. This is why (in contrast to Hofer et al.) we refrain from using the latent space as a basis for a classification task as it is not clear why the preserved data space topology would be beneficial for classification (as opposed to e.g. disentangled latent representations). We make this distinction now more clear in the paper to reduce confusion.
>
> > Computational issues of the $d$-dimensional Vietoris--Rips computation
>
> Thank you for raising this important point. This is a general limitation of persistent homology computations at the moment; it is not a specific limitation of our approach. However, we expect that forthcoming improvements in implementations (for example, GPU implementations or optimised implementations such as U. Bauer’s ‘Ripser’) or in approximation techniques will address this. As a side-note, we also want to mention that the question of the contribution of higher-dimensional features is not fully answered yet; some papers, such as Hofer et al. (2017) or ‘Learning metrics for persistence-based summaries and applications for graph classification’ by Zhao & Wang empirically found that higher-dimensional topological features are not required for tasks like graph classification.
>
> Please let us know if there are any other questions that we can answer for you concerning this paper.

---

### Public Comment · ~Aleks_Ivanov1 · 2019-11-08
**Missing citation**

A comment on missing citation. Since you are using the persistent homology invariants here is the reference for the paper where they were first introduced, under the name of canonical forms of filtered complexes: Barannikov, S.(1994). "Framed Morse complex and its invariants". Advances in Soviet Mathematics. 21: 93–115. Also the computations of the persistence diagrams are based on the algorithm described in section 2.1 in this paper.

---

> ### Author Response · Authors · 2019-11-08
> **re: Missing citation**
>
> Thanks for your comment! We added the citation, of which we were previously unaware, now in the revised version of the paper. It is really interesting to see how far the ideas of persistence go back!

---

### Decision · Program_Chairs · 2019-12-19

**Decision:**

Reject

**Comment:**

This paper introduces a new variant of autoencoders with an topological loss term.

The reviewers appreciated part of the paper and it is borderline. However, there are enough reservations to argue for it will be better for the paper to updated and submitted to next conference.

Rejection is recommended.